# The Value of Perinatal Factors, Blood Biomarkers and Microbiological Colonization Screening in Predicting Neonatal Sepsis

**DOI:** 10.3390/jcm11195837

**Published:** 2022-10-01

**Authors:** Isabel Cao, Norman Lippmann, Ulrich H. Thome

**Affiliations:** 1Divison of Neonatology, Center for Pediatric Research, University Hospital for Children, Liebigstraße 20a, 04103 Leipzig, Germany; 2Institute for Medical Microbiology and Virology, University of Leipzig, 04103 Leipzig, Germany

**Keywords:** neonatal sepsis, early-onset sepsis, late-onset sepsis, predictive factors, microbiological colonization screening, NICU

## Abstract

Background: Neonatal sepsis is one of the most important causes of elevated morbidity and mortality rates in neonatal intensive care units worldwide. While the clinical manifestations of neonatal sepsis tend to be nonspecific, its rapid development and life-threatening potential call for reliable markers for early detection. Methods: We conducted a retrospective single-center study including all neonates suspected of having developed neonatal sepsis from 2013 to 2016. Perinatal and clinical characteristics as well as microbiological and laboratory findings were evaluated. Neonatal sepsis was defined as either culture-proven sepsis (positive blood culture) or clinical sepsis (at least one symptom and elevated C-reactive protein (CRP) concentrations within 72 h with negative blood culture). We further differentiated between early-onset (EOS) and late-onset (LOS) sepsis. Results: Microbiological colonization screening by throat and rectal swabs frequently did not detect the organism that subsequently caused the sepsis. Depending on the age of the newborn with sepsis (EOS or LOS), associations between different anamnestic and clinical factors (prenatal or postnatal ones) were found. In particular, the central–peripheral temperature difference showed a strong association with LOS. Laboratory results useful for the early detection of neonatal sepsis included interleukin-6 (IL-6) and CRP concentrations. Conclusions: Elevated IL-6 >100 ng/L was a strong marker for neonatal sepsis. When choosing the antibiotics for treatment, data from microbiological colonization screening should be considered but not solely relied on. Some indicators of infection also depended on postnatal age.

## 1. Introduction

Neonatal sepsis remains one of the most important causes of elevated morbidity and mortality rates in neonatal intensive care units (NICU) all over the world [1,2,3]. In 2020, the World Health Organization estimated that 1.3–3.9 million newborns and up to 20 million children under the age of five suffer from sepsis worldwide every year [4]. 

Although neonatal sepsis commonly includes all cases of sepsis occurring during the neonatal period, there seem to be differences in the etiology and clinical features depending on the age of the newborn, dividing neonatal sepsis into early- (within the first 72 h of life) and late-onset sepsis (occurring after 72 h) [1]. It is therefore possible that different approaches are needed to detect early-onset sepsis (EOS) and late-onset sepsis (LOS).

Due to an immature immune system, especially in preterm neonates, newborns are particularly endangered by infectious agents [5]. Additionally, many preterm neonates need to remain in hospitals and NICUs for a long time, requiring intensive medical treatments and thus making them even more susceptible to sepsis. Neonatal sepsis can quickly become life-threatening [6]. It is therefore crucial to detect neonatal sepsis in the early stages and to start treatment early on. The clinical manifestations of neonatal sepsis, however, tend to be variable and nonspecific [6]. Hence, additional diagnostic means are needed to avoid underdiagnosis or overtreatment. The gold standard for detecting sepsis remains cultivating the infectious organism via blood culture, although this is known to have a limited yield. In particular, blood cultures obtained from newborns and small children show a low positivity rate, ranging from 7.4 to 12.8% in various studies [7,8,9,10]. 

Starting in 2007, the Commission for Hospital Hygiene and Infectious Disease Prevention at the Robert Koch Institute in Berlin, Germany (KRINKO), recommended a weekly colonization screening for the prevention of nosocomial infections in NICU patients with birth weights less than 1500 g [11]. This recommendation was subsequently updated in the years 2012 und 2013 and expanded to include all NICU patients [12,13]. Recommendations by this institution are considered to be standards of care and generally need to be followed in Germany. Consequently, weekly throat and rectal swabs were taken from all patients in the NICU at the University Hospital Leipzig. 

This study aimed to evaluate perinatal, clinical and blood sepsis indicators that might help verify or dismiss a suspected diagnosis of neonatal sepsis as early as possible based on information available at the time or shortly after the moment in which neonatal sepsis is suspected. Furthermore, we wanted to know how useful the recommended colonization screening was for predicting the sepsis-causing organisms. 

We hypothesized that:Anamnestic findings are associated with the subsequent development of infection;The clinically measured central–peripheral temperature difference is an early symptom of oncoming infection;Rectal and throat swabs predict which bacteria are subsequently found in blood cultures;There are differences in predicting EOS and LOS.

As blood culture results are known to have high false-negative rates but are most commonly used to investigate neonatal sepsis in studies, we also compared the blood culture results with elevated concentrations of C-reactive protein (CRP) within 72 h, another common way to confirm neonatal infections in a clinical setting. We investigated early- and late-onset sepsis (EOS and LOS) separately to find possible differences between these subtypes of the disease. Furthermore, we evaluated how often bacteria detected in the routine colonization screening were subsequently found in the blood culture.

## 2. Materials and Methods

This retrospective single-center study was approved by the institutional review board of the medical faculty at the University of Leipzig, Germany. We retrospectively evaluated all neonates with a suspected episode of sepsis from whom blood cultures were obtained from 1 January 2013 to 31 December 2016 at the NICU of the University Hospital Leipzig in Germany. This included a total of 513 patients with 629 episodes of sepsis workup. Multiple suspected cases of sepsis in one infant were conservatively interpreted as the same episode if the obtained blood cultures had an interval of one month or less, otherwise they were counted as separate episodes. Positive blood culture results with different organisms were interpreted as different episodes regardless of the time interval [14]. 

Blood cultures were obtained by direct venipuncture, not through indwelling catheters. Furthermore, following the universal recommendations all infants received weekly colonization screenings for potentially pathogenic and antibiotic-resistant bacteria through throat swabs and rectal swabs starting on day 1 of life. Bacteria were analyzed by standard culture methods. All bacteria, excluding those of the physiological colonization, were interpreted as potentially pathogenic organisms and reported to the NICU. Those considered high-risk colonizers by the KRINKO (*Serratia marcescens*, *Pseudomonas aeruginosa*, *Enterobacter* spp., *Acinetobacter* spp., *Klebsiella pneumoniae*, *Staphylococcus aureus*, multidrug-resistant Gram-negative bacteria or methicillin-resistant *Staphylococcus aureus*) were reported with special emphasis because isolation precautions were to be taken in those cases.

In addition to the usual physiological data, all infants routinely received continuous measurements of the central–peripheral temperature difference [15]. The skin temperature was continuously measured and logged by two standard temperature probes of our monitoring system (Philips Healthcare, Böblingen, Germany) on the lower back and on the sole of one foot. A difference ≥ 2 °C for ≥ 4 h within 72 h was considered suspicious for sepsis and was generally followed by an appropriate workup. 

Chorioamnionitis was defined as an intrapartum maternal temperature > 38 °C, CRP > 10 mg/L, leukocytosis > 10^9^/L or the detection of IL-6 >10^5^ ng/L in the amniotic fluid. Other prenatal factors included the preterm rupture of membranes (PROM: >24 h or > 7 d [16]), the type of delivery (caesarean section or vaginal), the maternal age at delivery, multiple pregnancy and anomalies in the amniotic fluid (color or odor).

The evaluated laboratory results included the IL-6 concentration, the CRP concentration at the time of the sepsis workup, the maximal CRP concentration within 72 h of sepsis workup, the neutrophil count (abs.), the bilirubin concentration and the lactate concentration.

Sepsis episodes were classified as early-onset sepsis (EOS, within the first 72 h of birth) and late-onset sepsis (LOS, older than 72 h).

Furthermore, we differentiated between culture-positive sepsis (CP), when a positive blood culture was detected, and clinical sepsis. The latter was recognized when at least one symptom occurred and the CRP concentration increased to values > 10 mg/L within 72 h of the sepsis workup at the time of the onset of symptoms. [17,18]. Cases without CRP measurements and neonates who underwent surgeries within 72 h were excluded.

The statistical analyses were performed using chi-square tests as well as univariate and multivariate logistic regressions. Cut-off values were determined using ROC analyses. The significance level was set at *p* ≤ 0.05. SPSS 25 software was used.

## 3. Results

We studied 629 suspected episodes of sepsis in all, including 263 cases in females and 366 cases in male patients. In total, 375 were EOS cases and 254 were LOS cases. The median gestational age was 31 + 6/7 weeks, ranging from 23 + 2/7 to 42 completed weeks. The median birth weight was 1.73 kg, ranging from 0.38 kg to 5.29 kg. Of 629 blood cultures, 79 were positive, yielding the following organisms (in decreasing order of frequency): coagulase-negative staphylococci (CoNS) in 47 cases (59.4%), Gram-negative bacteria in 20 cases (25.3%) and *Staphylococcus aureus* in 10 cases (12.6%). Other organisms, including the much-feared *Streptococcus agalactiae* (group B strep., two cases, 2.5%) were found much less frequently. There were three blood cultures in which more than one organism was detected. 

In total, 488 of the 513 patients with suspected neonatal sepsis were discharged alive.

The condensed anamnestic and clinical data and results are shown in Table 1. The birth weight, gestational age, presence of chorioamnionitis and persistent pulmonary hypertension were significantly associated with EOS, while an increased central–peripheral temperature difference was significantly associated with LOS. Respiratory distress syndrome was negatively associated with EOS.

### 3.1. Laboratory Results

Elevated IL-6 concentrations showed associations with all types of neonatal sepsis. IL-6 concentrations above the cut-off value of 100 ng/L showed statistically highly significant associations with neonatal sepsis in chi-square tests: CP EOS (OR = 6.307, 95% CI: 1.314–29.666, *p* = 0.011), clinical EOS (OR = 7.093, 95% CI: 4.238–11.873, *p* < 0.001), CP LOS (OR = 14.37, 95% CI: 6.095–33.881, *p* < 0.001) and clinical LOS (OR = 33.526, 95% CI: 12.521–89.770, *p* < 0.001).

ROC curves are shown in Figure 1, Figure 2, Figure 3 and Figure 4. All cut-off values and their specificity and sensitivity values were determined through the ROC analysis.

An elevated CRP above 5.5 mg/L at the time of sepsis workup predicted CP EOS with a sensitivity of 80% and a specificity of 74%. This association was also found to be statistically significant using the χ^2^ test (OR = 11.319, 95% CI 2.361–54.257, *p* = 0.001). With a higher cut-off, the sensitivity decreased while the specificity increased. Using a cut-off value of 10 mg/L, the sensitivity dropped to 40% while the specificity only marginally increased to 81%, and there was no longer a significant difference using the χ^2^ test. With the cutoff set at even higher CRP concentrations >10 mg/L, the specificity in detecting any clinical sepsis (both EOS and LOS) by CRP concentrations >10 mg/L was 100%, given the definition of clinical sepsis in this work. The sensitivity values were 52% and 73%, respectively. It was, however, not possible to confirm a statistically significant association between the CRP concentration and CP LOS by the χ^2^ or Fisher’s exact tests. Thus, a cut-off value was not determined. The ROC curves are shown in Figure 5, Figure 6, Figure 7 and Figure 8.

Other laboratory results (the neutrophil count, bilirubin concentration and lactate concentration) were not significantly associated with neonatal sepsis.

### 3.2. Microbiological Findings

#### 3.2.1. Early-Onset Sepsis

From the 286 cases of suspected EOS in which both blood cultures and rectal swabs were obtained, there were 30 cases in which suspicious organisms were found in the rectal swab at the time of the sepsis workup, but in only 11 cases an organism was also found in the blood culture. Of those 11 cases, there was only 1 case in which the same organism was found in the rectal swab and the blood culture (ESBL-producing *E. coli*). In another case, a different potential pathogen was detected in the rectal swab, and in 9 out of 11 cases with bacteremia there were no suspicious bacteria detected in the rectal swab beforehand.

The organisms most often found in the 11 blood cultures of EOS cases were *E. coli* and CoNS, which were found in 5 cases (45.5%) each. In one case, *Listeria monocytogenes* was detected.

Out of the 145 cases with suspected EOS in which both blood cultures and throat swabs were obtained, there were 19 cases in which suspicious organisms were found in the throat swab. By blood culture, however, a potential pathogen was detected in only 3 of the 145 cases. Of those three cases, there was only one case of EOS in which the same organism was found in the throat swab and the blood culture (*E. coli*). Blood-culture-positive infants tended to have potentially pathogenic organisms in their throat swabs more often, but this was not the case for their rectal swabs (Table 2).

#### 3.2.2. Late-Onset Sepsis

In 68 cases with suspected LOS, organisms were found in the blood culture. The detected bacteria were predominantly CoNS, which were found in 41 cases (60.3%), followed by Gram-negative bacteria in 15 cases (22.1%), *Staphylococcus aureus* in 10 cases (14.7%) and both *Streptococcus agalactiae* and *Enterococcus faecalis* in 2 cases each (2.9%).

There were 201 cases of suspected LOS in which both a blood culture and a rectal swab were obtained. In 51 of the 201 cases, organisms were found in the blood culture. Of those 51 cases, there were 10 in which, aside from a positive blood culture, organisms were detected in the rectal swab beforehand. However, we found the same organism in both the blood culture and rectal swab (*E. coli*) in only one of these ten cases.

In 218 cases with suspected LOS for which both blood cultures and throat swabs were obtained, 148 cases had potentially pathogenic organisms in the throat swab reported back before the time of sepsis workup. However, in only 56 of those 218 cases, an organism was detected in the blood culture. Among those 56 cases, there were 41 cases with suspicious organisms found in the throat swab beforehand, and in 17 of these 41 cases (41.5%) the same organism was found in the throat swab and the subsequent blood culture. Blood-culture-positive infants tended to have potentially pathogenic organisms in their throat swabs more often, but this was not the case for their rectal swabs (Table 2).

### 3.3. Organisms Identified in Swabs

The organisms identified in rectal swabs are shown in Table 3, and those identified in throat swabs are shown in Table 4. Overall, more isolates were found in the most immature infants, especially in infants who had received a sepsis diagnosis. Furthermore, throat-swab microbes were more diverse than rectal-swab microbes.

## 4. Discussion

In this work, we confirmed the diagnostic audit of common perinatal and clinical findings as well as laboratory test results for the early detection of neonatal sepsis. Important perinatal factors included gestational age, birthweight and the presence of chorioamnionitis. Clinically, an elevated central–peripheral temperature difference and pulmonary hypertension as well as increased IL-6 and CRP concentrations in blood samples were indicative of neonatal sepsis.

We further collected data on rectal and throat swabs and analyzed whether the sepsis-causing bacteria were detected in these. In EOS, neither rectal nor throat swabs were helpful. In LOS, there was only one case in which the rectal swab found the same organism as the blood culture. Throat swabs were slightly better, with a detection rate of 42%.

### 4.1. Perinatal and Clinical Characteristics

Prenatal factors (i.e., gestational age, birth weight and chorioamnionitis) showed the strongest statistical associations with EOS, as has been described previously [5,19]. We further detected a positive correlation between EOS and persistent pulmonary hypertension (PPHN), although this finding alone cannot answer the question of whether PPHN is a risk factor for EOS or vice versa. An inverse relationship was seen between respiratory distress syndrome (RDS) and EOS, which seemed counterintuitive. It is possible that similarities in clinical features lead to patients with RDS being more often suspected of having EOS and receiving a (negative) sepsis workup, while other patients without RDS were only scrutinized if they developed sepsis symptoms. Therefore, RDS should not be interpreted as a protective factor against EOS. Another surprising finding was that, among infants admitted to the NICU, smaller and more immature infants had EOS less often than larger, more mature infants. This seems counterintuitive at first glance. However, the respiratory symptoms are most likely due to RDS in smaller, more immature infants and due to EOS in older, more mature infants. Thus, small and very premature infants are often admitted because of immaturity, including RDS, while for larger, more mature infants, the reason for admission is the infection, leading to a selection bias. Other studies showed similar contradicting results, with birth weight and gestation age sometimes having both positive and negative associations with EOS [20,21,22].

In contrast to EOS, we found more postnatal factors showing significant associations with LOS, including a longer previous duration of hospitalization in the NICU. Another strong association was the one between LOS and an elevated central–peripheral temperature difference. The physiological basis of the central–peripheral temperature difference is the deterioration of the microcirculation in oncoming infection, which literally leads to “cold feet”, even inside an incubator [15]. We saw similar results as in other studies, including those conducted at our NICU [15,23,24]. Van de Puttelaar, on the other hand, found no significant link between LOS and central–peripheral temperature differences [J. L. van de Puttelaar, doctoral thesis, Utrecht, Netherlands, 2014]. It has to be noted, however, that the temperature measurements were taken at different sites: the central skin temperature was measured rectally or axillary, and the peripheral skin temperature was measured in the diaper, which would still be considered central by other authors. In contrast, we (and the other studies cited above) determined a difference between the central skin temperature (at the lower trunk) and the peripheral skin temperature (foot sole). Different temperature probe positions are the most likely explanation for the divergent results.

Some prenatal factors showed statistically significant associations with LOS. Surprisingly, infants resulting from single births tended to have a higher probability to develop LOS in comparison to those resulting from multiple births. This may be explained by our patient selection. To be included here, an infant needed to have a sepsis workup, and thus some symptoms, such as respiratory distress. Neonates resulting from multiple births may have respiratory distress more frequently and for other reasons aside from infection. Therefore, infants from multiple births may have a higher likelihood of receiving a sepsis workup without actually having an infection. Another rather surprising finding was a negative association between LOS and PROM > 24 h, which might be a chance result but also might be a result of the antibiotic treatment all pregnant women with PROM receive at our hospital.

### 4.2. Laboratory Findings

The analysis of IL-6 concentrations showed a strong predictive value for neonatal sepsis (both EOS and LOS). Our data favor a cut-off value at 100 ng/L, while different cut-off values have been suggested by others [25,26,27,28,29,30]. The sensitivity and specificity at this cut-off value showed similar results to those published in 2018 by the Association of the Scientific Medical Societies in Germany (AWMF) [6,26,31,32]. With higher cut-off values, specificity can be increased at the expense of sensitivity. Similar to other works [26,29,33,34], we found the IL-6 concentration to be a reliable and early-reacting parameter for diagnosing neonatal sepsis.

An elevated CRP concentration at the time of sepsis workup predicted EOS and LOS with limited sensitivity but good specificity values in this study. To elevate the sensitivity, a lower cut-off value would be needed, and this would lower the specificity. A cut-off of 10 mg/L is often taken as a positive CRP because it strikes a balance between an acceptable sensitivity and specificity. However, as the CRP elevation is known to be delayed for a considerable time, the sensitivity of the initial CRP concentration will probably never be high enough to suffice as a sole parameter for detecting neonatal sepsis and needs to be combined with other parameters, such as IL-6 [17].

For confirming the diagnosis of sepsis, we have two unsatisfactory choices. If we take only positive blood cultures as confirmatory for an infection, we will have a lot of truly infected infants in the “non-infected” group because of false-negative blood cultures. Furthermore, detection is delayed by several days, so treatment must be initiated solely by suspicion. On the other hand, if we accept additional biomarkers, other inflammatory diseases might be diagnosed as infection. Both choices will bias the results in one way or another.

We and others have found that a CRP increase 1–3 days after the onset of symptoms has a high specificity for infections in neonates [17,18]. We therefore defined clinical sepsis as a symptomatic episode with elevated CRP 1–3 days later. A negative CRP at that time was viewed as a ruled-out infection. This repeat CRP test at day 1–3 after the onset of symptoms (and treatment) is clearly distinguished from the initial CRP determined at the onset of symptoms, as shown in Figure 5, Figure 6, Figure 7 and Figure 8, where it was used along with IL-6 in the search for a possible infection. Since the CRP increase is rather slow, it may or may not be elevated in the initial measurement, even in cases with a positive blood culture.

Other causes of inflammation rarely play a part. Infants with chronic inflammation are easily distinguished because in these infants the CRP would not become negative after a week of antibiotic treatment. Such infants would have been excluded from our study.

The lack of associations of early CRP values with CP LOS stands in contrast to previous findings. National guidelines, concluded on the basis of numerous studies, showed a sensitivity of 46% and specificity of 86% for a cut-off value of 10 mg/L for predicting neonatal sepsis [6,17,18,35,36,37,38,39,40,41,42]. Ng et al. found a sensitivity of 46% and a specificity of 96% with a cut-off value of 12 mg/L for LOS in VLBW neonates [34]. The low number of patients with positive blood cultures and the known slow increase in CRP at the onset of infection might explain why the expected association was not found in our data. When examining the maximal CRP concentration within 72 h of the time of the sepsis workup (clinical sepsis), no clear associations with CP sepsis, either EOS or LOS, were found. This finding, however, is not sufficient to dismiss the CRP within 72 h as a confirmatory parameter for an infection because a negative blood culture does not mean that there was no bacteremia. Detecting organisms in blood cultures depends on many factors, including the sample volume, which always tends to be precarious in tiny infants. Further studies showed that repeatedly negative CRP (<10 mg/L) results after 24–48 h exclude an infection with high certainty [6,17,18,36,37,38,39,43], while on the other hand, it has been estimated that a bacteremia can be proven by bacteriological cultures in 10% of symptomatic neonates [6].

The bilirubin concentration plays a significant role in the diagnosis of adult sepsis and is part of the sepsis-related organ failure assessment score (SOFA) [44,45]. In neonates, bilirubin is often physiologically elevated, sometimes leading to neonatal jaundice with the need for phototherapy. We found no evidence for an association between the bilirubin concentration and both early- and late-onset neonatal sepsis. Likewise, the lactate concentration is an important factor in adult sepsis but was not statistically associated with sepsis in the studied infants.

### 4.3. Microbiological Findings

Microbiological screening in a NICU is generally aimed at preventing high-risk bacterial outbreaks by taking isolation precautions for infants identified as carriers of potentially problematic strains. Furthermore, the screening is thought to help orient the empirical treatment of nosocomial infections. In contrast, a microbiological diagnosis in sepsis is achieved by isolating an infectious agent from a usually sterile body site. In this work, we show that bacteria found in screening swabs have to be accounted for when treating neonatal infections, as the causative organism may be among them. However, in most cases, different organisms grew from the blood cultures. Therefore, the selection of antibiotics must consider the organisms detected in the screening as well as the possibility that other organisms may be causing the infection.

Especially in suspected EOS, very few patients had indicatory bacterial findings in the rectal and throat swabs before the time of the sepsis workup, probably because in those cases the patients had only a maximum of 72 h of contact with the bacterial environment outside of the amniotic fluid. Colonization and infection merely developed in a parallel fashion, which made analyses of colonization less predictive. The low yield of positive blood cultures also added to the incomplete picture left by the bacterial analyses. Colonization screening was thus not particularly helpful in selecting appropriate antibiotics since the organism later detected in the blood culture was only found in two cases during the screening, one in a rectal swab and one in a throat swab.

In cases with suspected LOS, on the other hand, far more suspicious organisms were found via rectal and especially throat swabs before the time of the sepsis workup, probably because the patients had far more contact with the environment by that time and more time to develop their microbiota. Rectal swabs, however, were still not very predictive since the organisms found in the rectal swab and the blood culture aligned in only one of ten cases. Throat swabs seemed to be more helpful in LOS. The detected microbiological flora was much more diverse, and in 17 out of 41 of the examined cases we found the same bacteria in both the throat swab and the subsequent blood culture. The higher number of matches to the blood-culture-proven sepsis agent using throat swab in comparison to rectal swab may indicate that the bacteria colonizing the mouth and throat area might be of higher diversity and of higher importance in the pathophysiological mechanisms of developing neonatal sepsis.

Of course, it has to be considered that blood cultures may also detect other organisms than those causing the actual sepsis because of contamination. Due to the small numbers, no further information in this particular cohort were available.

For selecting antibiotics for treatment, it seems sensible to consider the colonizing bacteria and their antibiogram but to also keep other typical yet undetected pathogens in mind. These findings align with the results of other research works in which the microbiological colonization screening was found to be a useful tool to analyze, contain and isolate multiresistant germs but was not found to be particularly precise in predicting neonatal sepsis and its microbiological cause [46,47,48,49,50].

### 4.4. Limitations of the Study

We conducted a retrospective study; therefore, the quality of the data depends on the correct and complete documentation of the patients beforehand. The cohort consists of all patients at the NICU within a certain time frame that were suspected of having developed sepsis. This leads to a lack of comparability to the overall collective, including those patients who were never suspected of having sepsis. As we conducted a single-center study, the number of cases was limited, and some local factors may have played a part. Another limitation of the study might be the unequal numbers in some of the examined subgroups and the low yield of the positive blood cultures, e.g., the EOS group, where there were only 11 cases with organisms found in the blood cultures, while the other 364 cases remained sterile. These imbalances might distort the statistical results.

The physicians were not blinded but were aware of the results of the rectal and throat swabs. However, positive swabs were never an indication to initiate treatment. Rather, symptoms and blood results were necessary by unit standards. Thus, bias from not blinding the swab results is unlikely. Furthermore, we do not have good data on cerebrospinal fluid (CSF) involvement because we generally do not perform lumbar punctures (LP) during the initial sepsis workup. We obtain a blood culture and start treatment without further delay. Any LP is performed later, when the infant is stabilized, and only in cases with symptoms of central nervous system involvement or with the growth of bacteria in the blood culture that are known to infiltrate the meninges, such as group B streptococci or Gram-negative rods. Since antibiotic treatment has already been started, bacteria are rarely found in the CSF. We then rely on the white cell count in the CSF to decide on adjustments in the dosage and treatment duration. Thus, no sufficient data on CSF involvement can be presented.

Furthermore, the alignment of the microbes found in the screening and blood cultures was performed only by genus and resistance testing, not genetics, which was not feasible in this retrospective study.

Finally, we tested a high number of hypotheses based on the same cohort and the same data to gain information about a variety of factors that might be associated with neonatal sepsis. The multiple comparison problem may lead to an α-error accumulation; therefore, the probability of false-positive errors is increased.

### 4.5. Future Outlook

Neonatal sepsis is a complex topic with high clinical relevance that makes ongoing research to improve the diagnostic tools, as well as therapeutic consequences, necessary. It is well-established that the outcomes of complex diseases can be improved through the standardization of procedures and clear recommendations [51,52,53]. Scores such as the nSOFA score, developed in 2020, are being developed and tested [54]. Some of them, such as the “Pediatric Sepsis Biomarker Risk Model (PERSEVERE)” include numerous new biomarkers [55,56]. Finally, new technologies, such as the genetic detection of organisms in the bloodstream may enable a deeper understanding of how, when and which bacterial colonizers become pathogenic.

## 5. Conclusions

We found various anamnestic and clinical factors that were associated with neonatal sepsis. Depending on the type of neonatal sepsis (EOS or LOS), the relevant factors seemed to originate either in the prenatal or postnatal period. A strong association was confirmed between central–peripheral temperature differences and LOS. Laboratory results useful for detecting neonatal sepsis included the concentrations of IL-6 and CRP. Evaluations are hampered by the low sensitivity of blood cultures. Biomarkers commonly used in adult sepsis scores, such as bilirubin and lactate, have no value in neonatal sepsis.

Since throat swab organisms are more diverse, throat swabs are the preferred method for microbiological screening surveillance, even though taking them seems to be less comfortable for the patient in comparison to rectal swabs. However, both throat and rectal swabs frequently did not detect the organisms that caused the sepsis. This is very important because, consequently, antibiotic treatments have to consider the whole spectrum of possible pathogens and cannot be limited to efficacy against the bacteria detected during the screening tests.

## Figures and Tables

**Figure 1 jcm-11-05837-f001:**
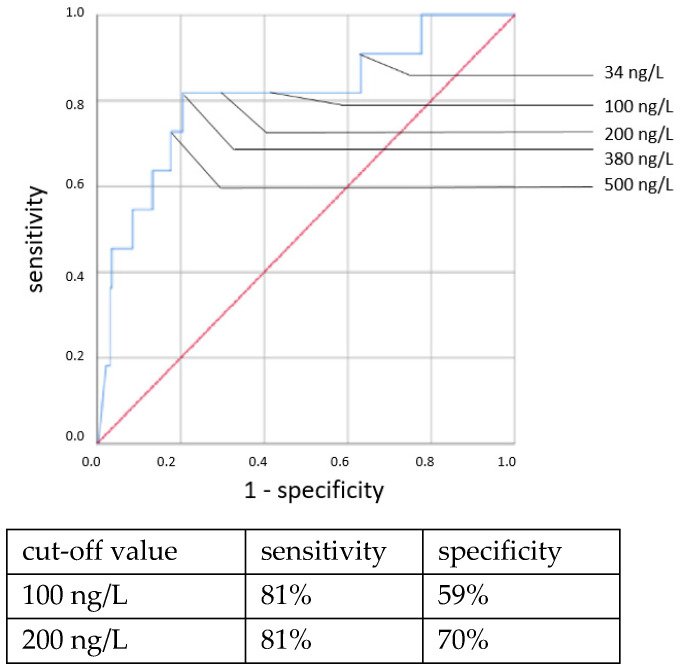
IL-6 concentration. Culture-positive EOS.

**Figure 2 jcm-11-05837-f002:**
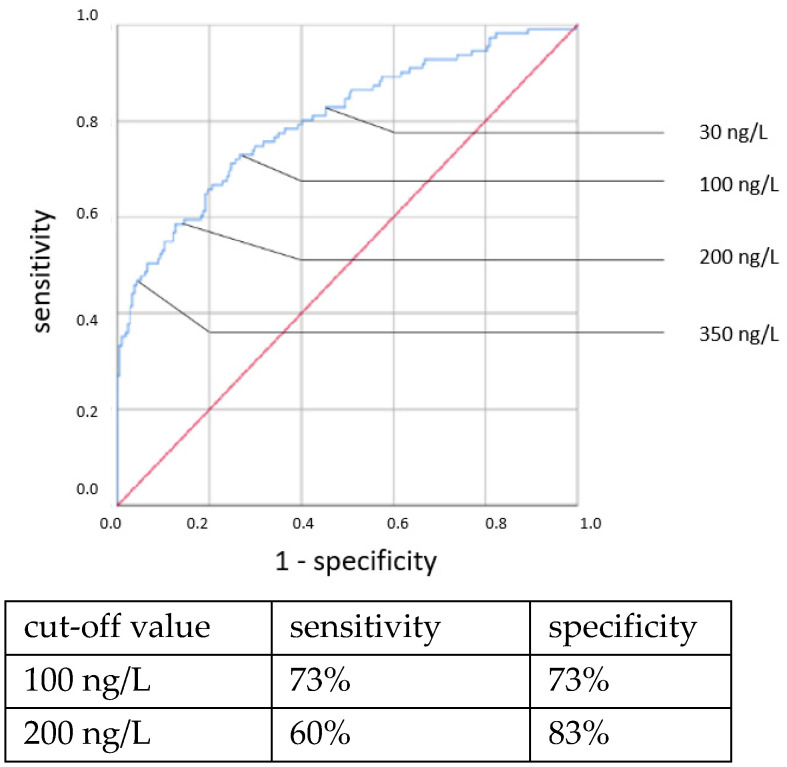
IL-6 concentration. Clinical EOS.

**Figure 3 jcm-11-05837-f003:**
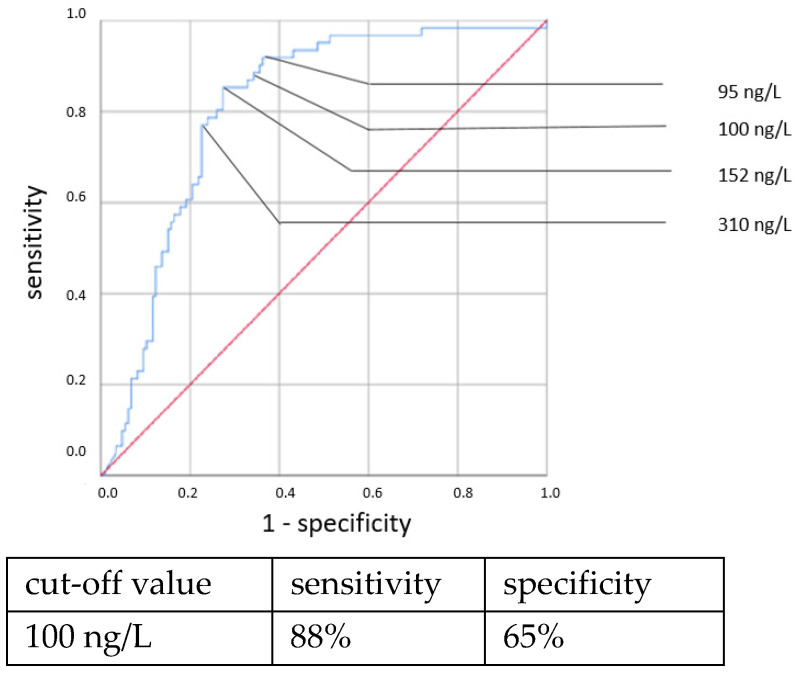
IL-6 concentration. Culture-positive LOS.

**Figure 4 jcm-11-05837-f004:**
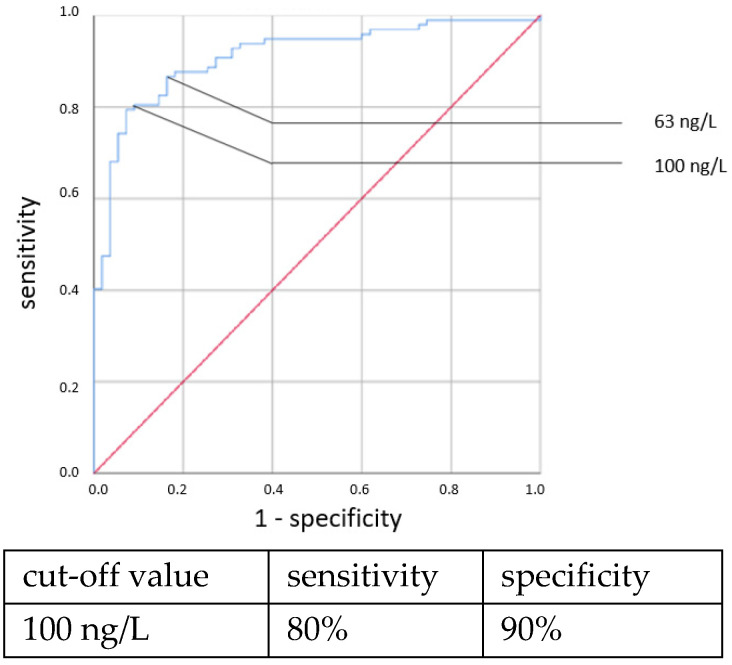
IL-6 concentration. Clinical LOS.

**Figure 5 jcm-11-05837-f005:**
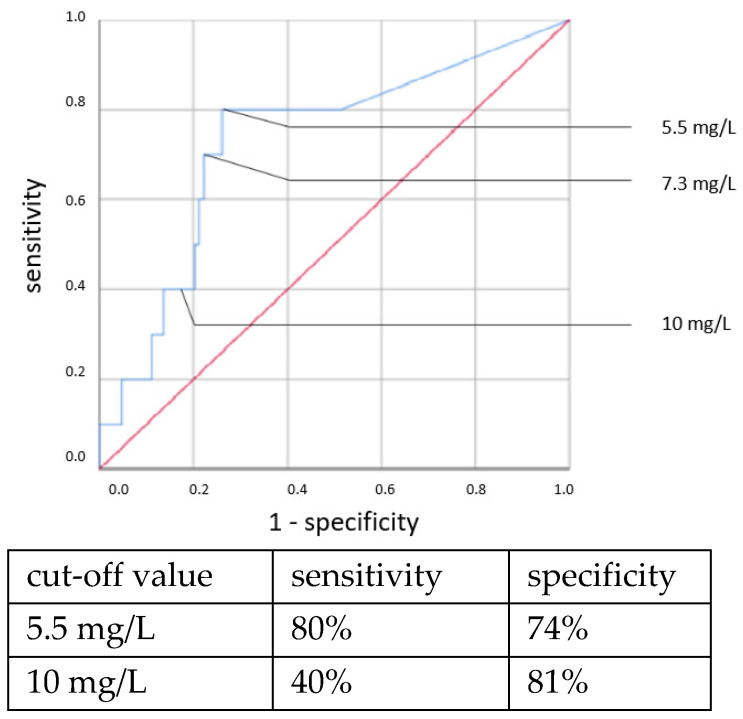
CRP concentration at the time of sepsis workup. Culture-positive EOS.

**Figure 6 jcm-11-05837-f006:**
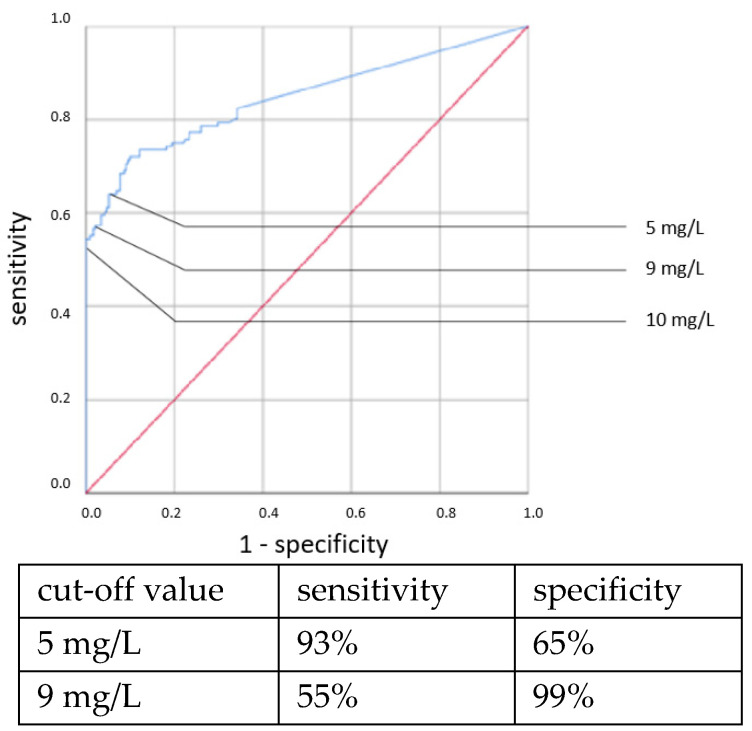
CRP concentration at the time of sepsis workup. Clinical EOS.

**Figure 7 jcm-11-05837-f007:**
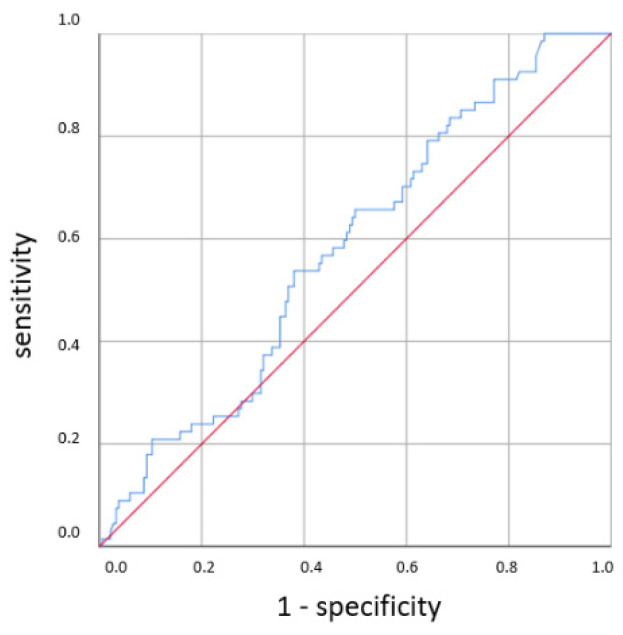
CRP concentration at the time of sepsis workup. Culture-positive LOS.

**Figure 8 jcm-11-05837-f008:**
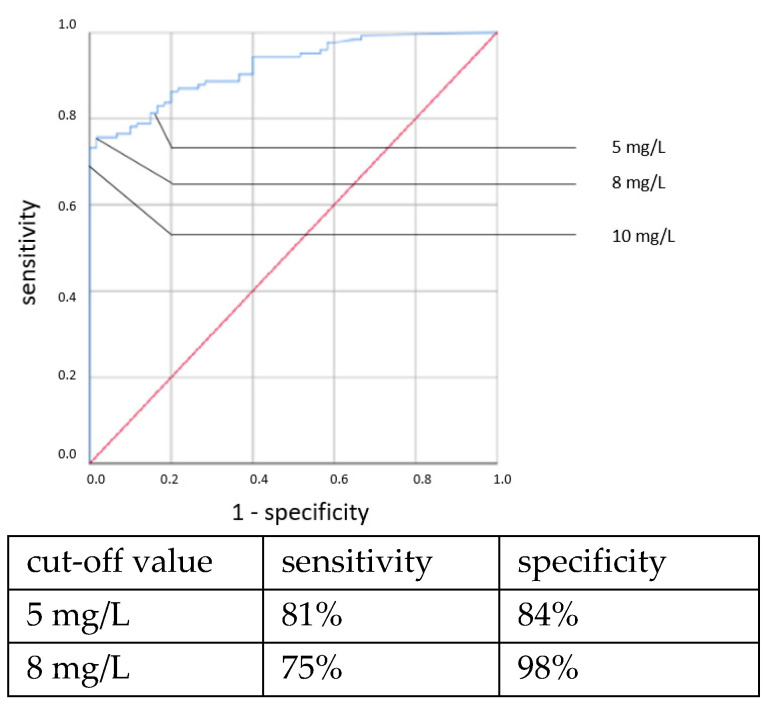
CRP concentration at the time of sepsis workup. Clinical LOS.

**Table 1 jcm-11-05837-t001:** Associations between anamnestic and clinical data and occurrence of infection episodes. Chi-square tests or logistic regressions were used to detect possible linkages to the sepsis workup. Significant results are shown in bold. Significant factors in the multivariate regressions are also underlined.

	EOS 152/374 (41%)				LOS 180/251 (72%)			
	N (EOS) *	N (no EOS) *	OR (CI)	*p*	N (LOS) *	N (no LOS) *	OR (CI)	*p*
Gestational age at birth (wks)	37 3/7 (23 3/7–42 0/7)	32 5/7 (23 5/7–41 6/7)	**1.077 (1.035–1.120)**	**<0.001**	28 1/7 (23 2/7–41 2/7)	28 2/7 (23 3/7–41 4/7)	1.029 (0.975–1.087)	0.299
Birth weight (kg)	2.72 (0.54–5.29)	1.98 (0.40–4.35)	**1.447 (1.196–1.752)**	**<0.001**	1.17 (0.38–4.02)	0.89 (0.40–3.56)	1.15 (0.828–1.598)	0.405
Male	87 (57%)	123 (55%)	1.168 (0.769–1.775)	0.465	107 (59%)	42 (59%)	1.012 (0.579–1.770)	0.966
1 min APGAR-score	7 (0–10)	7 (1–10)	0.984 (0.891–1.085)	0.741	7 (1–10)	7 (1–9)	0.973 (0.841–1.124)	0.708
5 min APGAR-score	8 (2–10)	8 (3–10)	0.997 (0.868–1.147)	0.971	8 (1–10)	7 (2–10)	1.046 (0.876–1.249)	0.621
Postnatal age at time of sepsis workup	†	†	†	†	22 (4–117)	24 (4–86)	0.999 (0.986–1.011)	0.847
Previous duration of hospitalization in NICU	†	†	†	†	22 (2–117)	22 (0–86)	1.000 (0.987–1.012)	0.960
Respiratory distress syndrome	**83 (55%)**	**154 (69%)**	**0.531 (0.346–0.815)**	**0.004**	144 (80%)	59 (83%)	0.814 (0.396–1.672)	0.574
Persistent pulmonary hypertension	**29 (19%)**	**26 (12%)**	**1.777 (1.000–3.160)**	**0.048**	34 (19%)	14 (20%)	0.958 (0.474–1.897)	0.880
Chorioamnionitis	**45 (30%)**	**46 (21%)**	**1.728 (1.066–2.799)**	**0.026**	30 (17%)	14 (20%)	0.758 (0.370–1.555)	0.449
Preterm rupture of membranes > 24 h	17 (11%)	33 (15%)	0.750 (0.400–1.406)	0.368	**15 (8%)**	**12 (17%)**	**0.433 (0.190–0.986)**	**0.042**
Preterm rupture of membranes > 7 d	7 (5%)	14 (6%)	0.744 (0.293–1.893)	0.534	12 (7%)	5 (7%)	0.925 (0.313–2.741)	0.889
Type of delivery (vaginal)	71 (47%)	82 (37%)	1.497 (0.984–2.276)	0.059	46 (26%)	14 (20%)	1.373 (0.699–2.696)	0.356
Maternal age at delivery	30 (15–45)	30 (15–44)	0.990 (0.955–1.025)	0.562	30 (17–42)	30 (15–39)	1.029 (0.978–1.082)	0.271
Multiple pregnancy	**15 (10%)**	**40 (18%)**	**0.498 (0.264–0.939)**	**0.029**	30 ((17%)	19 (27%)	0.533 (0.276–1.030)	0.059
Anomalies in amniotic fluid	54 (36%)	61 (27%)	1.516 (0.965–2.382)	0.070	**60 (33%)**	**14 (20%)**	**2.041 (1.042–3.997)**	**0.035**
Central–peripheral temperature difference	9 (6%)	22 (10%)	0.606 (0.271–1.357)	0.219	**77 (43%)**	**10 (14%)**	**4.516 (2.175–9.379)**	**<0.001**

* Dichotomous variables are shown as number (%), while continuous and categorical variables are shown as median (minimum–maximum). † Deferred because not useful. CI = 95% confidence interval.

**Table 2 jcm-11-05837-t002:** Rates of detecting target organisms in screening swabs.

	Blood-Culture-Positive Sepsis	Clinical Sepsis	No Sepsis
Throat swabs			
All neonates	42/59 (71.2%)	95/158 (60.1%)	40/142 (28.2%)
EOS	1/3 (33.3%)	14/55 (25.4%)	4/82 (4.8%)
LOS	41/56 (73.2%)	81/103 (78.6%)	36/60 (60%)
Rectal swabs			
All neonates	12/62 (19.3%)	39/202 (38.2%)	31/219 (14.2%)
EOS	2/11 (18.2%)	10/106 (9.4%)	19/165 (11.5%)
LOS	10/51 (19.6%)	20/96 (20.8%)	12/54 (22.2%)

**Table 3 jcm-11-05837-t003:** Types of organisms isolated from rectal swabs. □ Isolated from infants without sepsis. ◊ Isolated from infants with clinical sepsis. ● Isolated from infants with blood-culture-proven sepsis.

Type	23–<28 wks	28–<32 wks	32–<36 wks	>=36 wks
*Acinetobacter baumanii*	◊ ●		□	◊
*Acinetobacter ursingii*				
*Citrobacter braakii*		◊		
*Citrobacter freundii*				
*Citrobacter koseri*				
*Enterobacter amnigenus*				
*Enterobacter cloacae*	□ ◊ ●	□		◊
*Escherichia coli*	□ ◊ ●	□ ◊ ●	□ ◊ ●	◊ ●
*Haemophilus influenzae*				
*Haemophilus parainfl.*				
*Hafnia alvei*				
*Klebsiella oxytoca*				
*Klebsiella pneumoniae*	□ ◊ ●	□	□	◊
*Morganella morganii*				
*Pantoea* spp.				
*Proteus vulgaris*				
*Pseudomonas aeruginosa*				
*Pseudomonas putida*				
*Serratia liquefaciens*				
*Serratia marcescens*				
*Stenotrophomonas malt.*				
*Bacillus cereus*				
*Enterococcus faecalis*		◊		
*Enterococcus faecium*		□		
*Listeria monocytogenes*				
*Staphylococcus aureus*				
*Staphylococcus epiderm.*				
*Streptococcus agalactiae*				
*Candida albicans*				
*Candida glabrata*				

**Table 4 jcm-11-05837-t004:** Types of organisms isolated from throat swabs. □ Isolated from infants without sepsis. ◊ Isolated from infants with clinical sepsis. ● Isolated from infants with blood-culture-proven sepsis.

Type	23–<28 wks	28–<32 wks	32–<36 wks	>=36 wks
*Acinetobacter baumanii*	◊ ●	□ ◊	□	□ ◊ ●
*Acinetobacter ursingii*	◊ ●	◊		□
*Citrobacter braakii*				
*Citrobacter freundii*	□ ●			
*Citrobacter koseri*	◊ ●			
*Enterobacter amnigenus*	◊			
*Enterobacter cloacae*	□ ◊ ●	□ ◊ ●	□	□ ◊
*Escherichia coli*	□ ◊ ●	□ ◊ ●	◊ ●	◊
*Haemophilus influenzae*	□		◊ ●	◊
*Haemophilus parainfl.*	□ ◊		◊ ●	□ ◊
*Hafnia alvei*	◊ ●			□
*Klebsiella oxytoca*	□ ◊ ●	◊ ●		◊
*Klebsiella pneumoniae*	□ ◊ ●	□ ◊ ●		◊
*Morganella morganii*	◊			
*Pantoea* spp.	◊ ●			
*Proteus vulgaris*			◊	
*Pseudomonas aeruginosa*				◊
*Pseudomonas putida*	◊	□	□	
*Serratia liquefaciens*	□			
*Serratia marcescens*	◊ ●			◊
*Stenotrophomonas malt.*	◊			□
*Bacillus cereus*	◊ ●			
*Enterococcus faecalis*				◊ ●
*Enterococcus faecium*				
*Listeria monocytogenes*				◊
*Staphylococcus aureus*	□ ◊	□ ◊ ●	□ ◊ ●	□ ◊
*Staphylococcus epiderm.*	●	◊		
*Streptococcus agalactiae*	◊	◊ ●		□ ◊
*Candida albicans*	□	◊ ●	◊ ●	◊ ●
*Candida glabrata*				◊ ●

## Data Availability

The data set is pseudonymized, but this may not completely exclude identification of individual patients. Therefore, in accordance with our IRB, the full data set cannot be made publicly available. Excerpts of the data set can be made available to individual researchers upon request.

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
