# Peer review of "The Value of Perinatal Factors, Blood Biomarkers and Microbiological Colonization Screening in Predicting Neonatal Sepsis"

_jcm, 2022, doi:10.3390/jcm11195837_

Round 1

Reviewer 1 Report

- line 87-88: "all infants routinely received continuous measurement of the central-peripheral temperature difference": please clarify the methodology used (it is better explained only in lines 242-244). 

- line 194: "did we find": change in "we found". 

- line 246-250: the concept is quite hard to understand, please rephrase or clarify. 

- line 369: "Furthermoe", please correct. 

Author Response

- line 87-88: "all infants routinely received continuous measurement of the central-peripheral temperature difference": please clarify the methodology used (it is better explained only in lines 242-244). 
Thank you for the thorough review. The infomation was added

- line 194: "did we find": change in "we found". 
The wording was corrrected, thank you

- line 246-250: the concept is quite hard to understand, please rephrase or clarify. 
We are sorry for the unclear wording and have revised it. 

- line 369: "Furthermoe", please correct. 
The typo was corrrected, thank you

Reviewer 2 Report

This retrospective work on the experience of a single NICU confirms the usefulness of the IL-6 assay in the diagnosis of neonatal  sepsis and the difficulty of a microbiological diagnosis, especially in early-onset neonatal sepsis .

I suggest to better define "clinical sepsis" (definition in lines 101-104) indicating the symptoms or any score considered.

Table 1 is difficult to read. What do the authors mean by the term "laboratory confirmed sepsis"?

 Did the authors assess procalcitonin levels?

I believe it is important to underline that the microbiological screening in a NICU is aimed at orienting the empirical treatment of nosocomial infections and preventing  high-risk bacterial outbreaks. Microbiological diagnosis in sepsis is achieved by isolating an infectious agent from a usually sterile body site. In this regard, was a lumbar puncture done in suspected cases? What was the empirical therapy of EOS and LOS? Was it influenced by the microbiological screening? Did the infectious screening lead to fungal isolations? A table with the isolated at different gestational ages by the weekly colonization screening would be useful.

Author Response

-This retrospective work on the experience of a single NICU confirms the usefulness of the IL-6 assay in the diagnosis of neonatal  sepsis and the difficulty of a microbiological diagnosis, especially in early-onset neonatal sepsis .
Answer: Yes, these are important conclusions from our work.

I suggest to better define "clinical sepsis" (definition in lines 101-104) indicating the symptoms or any score considered.
Answer: The wording was optimized. We did not use any scores. 

Table 1 is difficult to read. What do the authors mean by the term "laboratory confirmed sepsis"?
Answer: This is a good point. We meant what we otherwise termed "clinical sepsis". We habe replaced the wording. 

Did the authors assess procalcitonin levels?
Answer: this is an important point. However, Franz et al. had shown 1999 that "The combination of IL-8 and CRP is more reliable than PCT as a test for early diagnosis of BI [bloodstream infection] in newborn infants." Therefore the measurement of procalcitonin is not part of the sepsis workup in this institution, and thus, no procalcitonin data was available. 

I believe it is important to underline that the microbiological screening in a NICU is aimed at orienting the empirical treatment of nosocomial infections and preventing  high-risk bacterial outbreaks. Microbiological diagnosis in sepsis is achieved by isolating an infectious agent from a usually sterile body site. 
Answer: The reviewer is corrrect. We have added this information to the discussion.

In this regard, was a lumbar puncture done in suspected cases? 
Answer: this is a good point. However, lumbar punctures were rarely done at our institution. Furthermore, we generally do not perform LPs during the initial sepsis workup. We get a blood culture and start treatment without further delay. Any LP is done later, when the infant is stabilized. Since antibiotic treatment has already been started, bacteria are rarely found in the CSF obtained by the LP. Still, we can find the increase in WBC in the CSF which we deem sufficient for deciding on adjustments for dosage and treatment duration. For these reasons, we only have 8 LPs in our study group, which is insufficient to make well-founded statements. We have added this infomation to the discussion.

What was the empirical therapy of EOS and LOS? Was it influenced by the microbiological screening? 
Answer: The empirical therapy for EOS at that time was ampicillin and cefotaxim. For LOS, in preterm infants, it was cefotaxim and vancomycin. If there was screening result available, it was considered in the selection of sntibiotics. 

Did the infectious screening lead to fungal isolations? A table with the isolated at different gestational ages by the weekly colonization screening would be useful.
Answer: Tables 3 and 4 showing the isolates were added. There were 8 fungal isolates in throat swabs.

Reviewer 3 Report

This study was a retrospective review of neonates treated for sepsis at a single institution.  Multiple characteristics (IL-6, CRP, clinical features, maternal factors) were compared in neonates with clinical and culture-proven sepsis.  Elevated IL-6 and central-peripheral temperature differences were associated with sepsis.  Rectal and throat colonization did not seem to be associated with detected organisms.  

There are various minor grammatical and syntax errors, the manuscript would benefit from editing from a native English-speaker. 

It is unclear what their hypothesis is. 

Table 1 is very difficult to read due to the size of the font.  It is also unclear what the authors mean by microbiological LOS vs. laboratory confirmed LOS.  

There are two table 1s.  

Lines 153-165 needs to be written more clearly, the paragraph is very muddled and not well-organized.  

In section 3.2, how did colonization rates compare between infected vs. uninfected?  If rates were similar, it would not support their reported detection rate by throat swab as it could have happened by chance.  

Were physicians aware of results of the rectal and throat swabs?  Any data to suggest results biased a physician's decision to treat clinical sepsis?

How common and easy is it for labs to obtain IL-6?  At some institutions, it is not common practice to check IL-6, and I don't know the turnaround time for it at their institution.  Would results be returned in a timely fashion?  They discussed the importance of considering the timing of CRP, what was the average timing of their CRP and IL-6 levels?  And were either followed serially?

They introduced results on RDS, PPHN in the discussion (line 217) without any explanation of the data in the results section (asides from being in Table 1).  They need to spell these acronyms out, and also explain the data better in the results section.  Also, since the criteria/definitions used for RDS and PPHN can be debatable, they need to explain how these conditions were defined in patients (e.g. CXR, diagnosis code, echo?) in the Methods section.  The discussion in 4.1 is overall disorganized and I do not know what their main argument is.  

What is the significance of central-peripheral temperature difference?  Can they provide some background on the reason this is associated with sepsis for those not familiar?

An unpublished doctoral thesis (citation #23) needs to be cited differently.

Lines 246-251 is speculative and disorganized regarding multiples vs. singletons, needs to be reworked.

Line 272: "As clinical neonatal sepsis was determined by CRP...the analysis...might be prone to a statistic bias."  It's very likely that clinicians chose to treat clinical sepsis due to an elevated CRP, so this association is very biased.  This needs to be more clearly stated. If they argue that an elevated CRP can be indicative of infection despite negative cultures (lines 283-292), which is what I assume was how management was determined at their institution, then that is going to muddle their results.  Lines 292: "too many false negative blood cultures" and possible overtreatment because of elevated CRPs may have disturbed the analyses.

I am having a hard time keeping track of the different types of sepsis they describe, it is very confusing for readers and needs to be more clear in the manuscript.  The different types I found were "clinical sepsis," "microbiologically proven sepsis," "laboratory confirmed sepsis," "suspected sepsis," "culture-proven sepsis," and "sepsis" alone.  

The authors need to provide more rationale for the idea of swabbing to guide antibiotic coverage, their results are not convincing.  The flora on mucosal surfaces are diverse, but the organisms that cause neonatal sepsis are relatively limited.  Also, missing in their data is a comparison of flora diversity in non-infected neonates - their few cases of detecting organisms that were also in the blood could have been by chance alone unless they can demonstrate that these organisms were at least absent in non-septic neonates.  But again, hard to make the argument that the organisms detected on swabs were causative without more information (e.g. serotyping/genotyping).

Author Response

This study was a retrospective review of neonates treated for sepsis at a single institution.  Multiple characteristics (IL-6, CRP, clinical features, maternal factors) were compared in neonates with clinical and culture-proven sepsis.  Elevated IL-6 and central-peripheral temperature differences were associated with sepsis.  Rectal and throat colonization did not seem to be associated with detected organisms.  

There are various minor grammatical and syntax errors, the manuscript would benefit from editing from a native English-speaker. 
Answer: We have reviewed spelling and grammar and corrected all errors we could find. Since there was a tight deadline, we were unable to include an additional review by a native English speaker, because this would have required another week's time after finalization by the researchers.. We can do this if this is still needed and we get 1 week additional time. 

It is unclear what their hypothesis is. 
Answer: This is a good point. We have added our hypotheses.

Table 1 is very difficult to read due to the size of the font.  It is also unclear what the authors mean by microbiological LOS vs. laboratory confirmed LOS. 
Answer: We have increased the font size of the table, by setting this page into the Landscape format. Furthermore, we have revised the table according to suggestions by reviewer 4. In addition, we have reworked the sepsis definitions in order to ensure that it is always clear what is meant. 

There are two table 1s. 
Answer: The reviewer is correct. Ther should only be one Table 1. The other tables were part of the figure legends and no longer counted as separate tables.

Lines 153-165 needs to be written more clearly, the paragraph is very muddled and not well-organized. 
Answer: We have rewritten this paragraph 

In section 3.2, how did colonization rates compare between infected vs. uninfected?  If rates were similar, it would not support their reported detection rate by throat swab as it could have happened by chance. 
Answer: A new table 2 showing this information was added. Rates of colonization with target bacteria in the throat were higher in infected infants, so the hypothesis raised by the reviewer appears to be important but not applicable to our data. 

Were physicians aware of results of the rectal and throat swabs?  Any data to suggest results biased a physician's decision to treat clinical sepsis?
Answer: Yes, the physicians were aware of the results of the rectal and throat swabs. However, it was always clear for all team members that colonization is not infection, and that colonization was no reason to initiate treatment. We have added this information to the discussion section.

How common and easy is it for labs to obtain IL-6?  At some institutions, it is not common practice to check IL-6, and I don't know the turnaround time for it at their institution.  Would results be returned in a timely fashion?  They discussed the importance of considering the timing of CRP, what was the average timing of their CRP and IL-6 levels?  And were either followed serially?
Answer: This is an important point. We usually get the IL-6-concentration within 90 minutes ob blood sampling.

They introduced results on RDS, PPHN in the discussion (line 217) without any explanation of the data in the results section (asides from being in Table 1).  They need to spell these acronyms out, and also explain the data better in the results section.  Also, since the criteria/definitions used for RDS and PPHN can be debatable, they need to explain how these conditions were defined in patients (e.g. CXR, diagnosis code, echo?) in the Methods section.  The discussion in 4.1 is overall disorganized and I do not know what their main argument is.  
Answer: The abbreviations are now spelled-out and included in the Resuslts text. 4.1. was revised.

What is the significance of central-peripheral temperature difference?  Can they provide some background on the reason this is associated with sepsis for those not familiar?
Answer: This is an important point. We have previously published on this issue. (Ussat M et al., 2015, citation # 15). The physiological basis is the deterioration of the microcirculation in oncoming infection, which literally leads to "cold feet" even inside a heated incubator, which we can measure by an increasing temperature difference. 

An unpublished doctoral thesis (citation #23) needs to be cited differently.
Answer: The reviewer is correct. We have changed the citiation.

Lines 246-251 is speculative and disorganized regarding multiples vs. singletons, needs to be reworked.
Answer: The reviewer is correct. We have rewritten this paragraph.

Line 272: "As clinical neonatal sepsis was determined by CRP...the analysis...might be prone to a statistic bias."  It's very likely that clinicians chose to treat clinical sepsis due to an elevated CRP, so this association is very biased.  This needs to be more clearly stated. If they argue that an elevated CRP can be indicative of infection despite negative cultures (lines 283-292), which is what I assume was how management was determined at their institution, then that is going to muddle their results.  Lines 292: "too many false negative blood cultures" and possible overtreatment because of elevated CRPs may have disturbed the analyses.
Answer: The reviewer raises a difficult and important point. We have 2 unsatisfactory choices. If we take only positive blood cultures as confirmatory for an infection, we will have a lot of truly infected infants in the "non-infected" group, because of the falsely negative blood cultures, which not only occur at our institution. On the other hand, if we accept additional biomarkers, other inflammatory diseases might be diagnosed as infection. Both will bias the results in one way or another. 
We and others have found that a CRP increase 1-3 days after the onset of symptoms has a high specificity for infections in neonates. Other causes of inflammation rarely play a part. This has been laid out in multiple publications, and the most important ones are cited in our manuscript. There is, to our knowldge, no better biomarker for confirming an infection in a neonate. 
Therefore, we defined clinical sepsis as a symptomatic episode with elevated CRP 2 days later. A negative CRP at that time was viewed as ruled-out infection. We hope to prevent muddled results, since we clearly distinguished the repeat CRP test at day 1-3 after the onset of symptoms (and treatment) from the initial CRP determined at the onset of symptoms, as shown in Figs 5-8, where it was used along with IL-8 in search for a possible infection. Since the CRP increase is rather slow, it may or may not be elevated in the initial measurement, even in cases with a positive blood culture. 
Infants with chronic inflammation are easily distinguished, because in these infants, the CRP would not become negative after a week of antibiotic treatment. Such infants would have been excluded from our study. We have added these lines of thought outlined here to the discussion section. 

I am having a hard time keeping track of the different types of sepsis they describe, it is very confusing for readers and needs to be more clear in the manuscript.  The different types I found were "clinical sepsis," "microbiologically proven sepsis," "laboratory confirmed sepsis," "suspected sepsis," "culture-proven sepsis," and "sepsis" alone. 
Answer: The reviewer is correct. We revised the manuscript to use only "clinical sepsis" and "culture-proven sepsis". 

The authors need to provide more rationale for the idea of swabbing to guide antibiotic coverage, their results are not convincing.  The flora on mucosal surfaces are diverse, but the organisms that cause neonatal sepsis are relatively limited. Also, missing in their data is a comparison of flora diversity in non-infected neonates - their few cases of detecting organisms that were also in the blood could have been by chance alone unless they can demonstrate that these organisms were at least absent in non-septic neonates.  But again, hard to make the argument that the organisms detected on swabs were causative without more information (e.g. serotyping/genotyping).
Answer: The reviewer raise a good point. After several outbreaks of multi-resistant organisms in German ICUs (not only NICUs), regular swabbing has been recommended by the Robert-Koch Institute in Berlin (sort of equivalent to the CDC in the US) and thus become quasi-mandatory in Germany. The main idea was to prevent future outbreaks. However, we do not actually know how helpful the swabbing results are for clincal practice, which was one motivation to do this analysis. We have added information on swabbing and isolated organisms to the mnauscript.

We thank the reviewer for the most thorough review.

Author Response

See PDF
